# Piezoelectric Energy Harvesting from Rotational Motion to Power Industrial Maintenance Sensors

**DOI:** 10.3390/s22197449

**Published:** 2022-09-30

**Authors:** Jaakko Palosaari, Jari Juuti, Heli Jantunen

**Affiliations:** Unit of Microelectronics, University of Oulu, 90014 Oulu, Finland

**Keywords:** energy harvest, piezoelectric, rotational motion, softening effect, condition monitoring

## Abstract

In industry, forecasting machinery failures could save significant time and money if any maintenance breaks are predictable. The aim of this work was to develop an energy harvesting system which could, in theory, power condition monitoring sensors in heavy machinery. In this study, piezoelectric-cantilever-type energy harvesters were attached to a motor and spun around with different rotational speeds. A mass was placed on the tip of the cantilevers, which were mounted pointing inward toward the center axis of the motor. Pointing a cantilever tip inward and increasing the distance from the center axis of the motor decreased the natural resonance frequency significantly and thus enabled higher harvested energy levels with lower rotational frequencies. Motion of the cantilever was also controlled by altering the movement space of the tip mass. This created another possibility to control the cantilever dynamics and prevent overstressing of the piezoelectric material. Restricting the movement of the tip mass can also be used to harvest energy over a wider frequency range and prevent the harvester from getting trapped into a stagnant position. The highest calculated raw power of 579.2 µW at 7.4 Hz rotational frequency was measured from a cantilever with outer dimensions of 25 mm × 100 mm. Results suggest that an energy harvesting system with multiple cantilevers could be designed to replace batteries in condition sensors monitoring revolving machinery.

## 1. Introduction

Piezoelectric energy harvesting from mechanical stress has been widely studied over the last decade, as electronic devices have surrounded our daily lives. The goal has been to replace or make more sensor solutions that could either be used solely with harvested energy or with prolonged battery life. Many piezoelectric energy harvesters have been studied from vibrating sources, where a mass is used to tune a cantilever type harvester to a certain frequency. Refs. [1,2,3,4,5,6] show that, unfortunately, in a large number of cases, the vibrating sources in nature are at a very low frequency and require a large mass proportional to the size of the actual energy harvesting part. Mass tuning to lower the resonance can vastly improve the harvester efficiency, but the corresponding stress build-up in the piezoelectric material increases the risk of fatigue failure [7].

The same dilemma comes with harvesting from rotational motion. Many studies have used a magnet to provide the desired force to bend a cantilever with each rotation [8,9]. Sometimes, it is not possible to use magnets, or they do not create enough force to bend the piezoelectric cantilever, resulting in an insufficient amount of energy to provide the required power; for example, to power a sensor’s electronics. Another common approach to convert revolving energy into vibration is to attach the piezoelectric cantilever through tooth-like parts of the rotating body. Tien et al. in [10] designed a system, where wind drives rotating blades and teeth on the body move the piezoelectric cantilevers to generate electrical energy. A novel technique was introduced by Yang et al. in [11], where wind energy drives a body with piezoelectric cantilevers inside a chamber together with bouncing steel balls. Balls bounce around the chamber and hit the cantilevers to create electrical energy. Other articles [12,13] have relied on very flexible beams, which in theory could stiffen due to centrifugal force as the rotation speed increases and hit a resonance at the desired rotation speed. Demanding design parameters and a quite high and narrow resonance band might be a challenge with this approach. Over the past few years, more articles [14,15,16,17,18,19] have been published about piezoelectric energy harvesting from a rotational motion utilizing the centrifugal softening effect. The softening effect can be used to lower the natural resonance frequency by pointing a cantilever-type harvester toward the center axis.

In this paper, the focus is on harvesting energy from rotational motion with a piezoelectric bimorph type cantilever utilizing the softening effect. In addition, it is shown how limiting the tip movement space and distance from the center axis affects the usable frequency range, resonance and the amount of energy being harvested. This energy could be used for sensing tires or heavy machinery with rotational parts, even with very low 60–600 rpm rotational speeds. Harvested energy could, for example, be used to power tire pressure or a wheel motion balance sensor to forecast tire wear or bearing failures. In industry, forecasting machinery failures could save significant time and money if any maintenance breaks are predictable. For instance, the conditions of large revolving machines, such as those in the paper industry, conveyor belts in production lines, or port-loading machinery, could be monitored. This study shows how positioning and alignment of the cantilever-type harvester play a crucial role. In particular, the tip motion control can be modified to enhance the harvested energy and make the design more robust.

## 2. Prototype Manufacturing and Test Setup

Piezoelectric ceramic and steel parts for prototypes were laser machined with a LPKF protolaser enabling high-precision parts. Three prototypes were manufactured in order to measure the harvested power as a function of the rotation frequency. The dimensions of the bimorph-type cantilevers and mounting holder can be seen in Figure 1. The lasered piezoelectric parts (PZ-5A) of 24 mm × 24 mm × 0.5 mm in size were installed with a conductive epoxy on both sides of the lasered steel beam (150 µm) for all prototypes. The free length from the clamp to the mass was 79 mm for each cantilever. Different tip masses (7.05 g/Ø 12 mm, 8.94 g/Ø 13 mm, and 13.75 g/Ø 15 mm) were tested for the three prototypes to tune the resonance of the harvester.

Clamping mechanics for the harvesters were made with a Markforged Mark Two 3D printer. Carbon-fiber-filled nylon ensured high-durability parts for mounting on a plate rotated by the motor (Figure 2).

Prototypes were attached on a round aluminum plate that was clamped onto the revolving motor axle (Figure 3 and Figure 4). The voltage from the harvester was measured through a rectifier, over an electrical load, sent wirelessly and recorded with analyzing software. The sample rate of the wireless measurement device was 300 Hz. Slow-motion videos were used to further understand the mechanics of operation (Appendix A).

The first test sample was made to find the correct orientation of the cantilever suitable for energy harvesting at low frequency rotation speeds. It was noted that the centrifugal force increased as a function of rotating speed and as the distance to the center axis. Gravitational force dominated at very low rotation speeds and shifted toward the cantilever according to alignment but was overpowered by the centrifugal force as the rotation speed increased. The clamping position of the piezoelectric cantilever was chosen so that the long side of the cantilever harvester was aligned with the rotating plane and the shorter side was perpendicular to the plane. In this way, the centrifugal force either stretches or compresses the harvesting ceramic and only starts to bend the beam after gravity has deflected it from its straight position. More importantly, in this alignment, the gravitational force shifts dominate at lower rotation speeds and bend the beam and piezoelectric material (Figure 3).

The energy harvesting circuit can be seen in Figure 5. The harvester voltage was measured through a rectifier and over a load of 110 kOhm. This electrical load was determined by the wireless voltage measurement device and could not be optimized for harvesting frequency/rotation speed or for the capacitance of the piezoelectric parts. Due to this disadvantage, the harvested energy was also measured by loading a capacitor over time and subsequently comparing this to the results from the previous method.

Energy was calculated from the following equation:*E* = 0.5 × *C_e_* × *V*^²^(1)
where *C_e_* is the capacitance of the capacitor, and *V* is the voltage level charged into the capacitor. This was compared to the calculated energy: *E* = *P* × *t*(2)
*P* = *V*^²^/*R_L_*(3)
where *t* is time and *V* is the measured voltage over an electrical load *R_L_*.

## 3. Results and Discussions

The prototype with the heaviest mass produced the highest energy and was measured more accurately with different restrictions of the tip mass movement. Figure 6 shows how the tip mass movement was limited and how the beam could bend with the highest (77 mm) and lowest limiters (48 mm) before making contact. Thus, the looseness of the tip movement decreased with the height of the limiter.

The measurement results (Figure 7) show the amount of harvested power at different rotational speeds and with six different movement limitations of the tip mass. This shows that four times more power could be harvested with the loosest movement limitation of the tip mass (48 mm) versus the tightest one (77 mm). However, with the loosest limitation, the mass became more easily pinned/trapped into a stagnant position (4.6 Hz) due to centrifugal force (“Appendix A”). The calculated ratio of the resonance’s center frequency to its half-power bandwidth showed that the widest frequency bands were with the tightest (70 mm) and the middle length limiter (60 mm), having Q factors of 5.65 and 5.80, respectively. The two limitations of 48 mm and 68 mm, which had the highest power levels, showed Q factors of 8.07 and 11.02, respectively. The highest calculated raw power of 208.8 µW at 4.6 Hz rotational frequency was measured with the 13.75 g/Ø 15 mm tip mass and 48 mm limitations. This computes to ~725 µW/cm³ for the piezoelectric material.

Essential parameters to raise harvester power were the rotational speed and amount of deviation of the cantilever tip. Higher rotational speed leads to a higher bending frequency, which determines how many times an electric charge is generated. Shorter limiters allow the cantilever tip to bend more, thus a higher stress is inflicted upon the clamped piezoelectric material, leading to the generation of a greater charge. For example, Figure 6 shows how longer limiters (77 mm) stop the cantilever free beam traveling ~17 mm, and shorter limiters (48 mm) allow the beam to bend ~39 mm before collision. In addition, collision between the limiter and the harvester leads to a wider frequency band; the same observation was also made by Rui et al. [19]. Additionally, more energy can be harvested due to shock excitation caused by the collision between the limiter and harvester. The frequency band of the harvested energy could be increased with a returning mechanism that bounced the mass back into motion when pinned/trapped into a stagnant position. See Appendix A to further understand the mechanics of operation.

### 3.1. Testing Different Harvesting Electronics

The previous measurements were made by converting the bipolar piezoelectric output into a unipolar voltage, which was measured over an electrical load to calculate harvested power (Figure 5). This is not the optimal electrical configuration to harvest all the available energy from the system. Although this investigation was devoted to studying and enhancing the mechanics behind the harvested energy, in the next measurements, the harvested energy was also measured by loading a capacitor over time and compared to the previous method (Figure 8).

The harvester with a 13.75 g tip mass was measured. Energy was harvested through a rectifier into a capacitor, and the voltage was measured from the capacitor after different time periods. This harvested energy into capacitor was compared to the calculated energy from the measured voltage over an electrical load at the same rotation frequency of 4.5 Hz as in previous measurements. Two different-sized capacitors of 103.7 µF and 448 µF were used for comparison, and the voltage was measured over time. The results can be seen in Figure 9.

Results show that at least ~70% more energy (39.78 mJ) could be harvested with an optimal capacitor loading (blue and red lines) over time than was calculated by measuring the voltage over an electrical load (23.35 mJ). Loading a capacitor is closer to the application where all the energy can be stored directly into the system. As stated earlier, the harvesting electronics used for the wireless voltage measurement is not an ideal case. There are many studies devoted to energy harvesting circuits, which can further improve the harvested energy amount significantly [20,21,22,23]. In the end, designing an optimum circuit for energy harvesting goes hand in hand with the mechanical design. The application determines the mechanical design, which in turn, sets the electromechanical properties such as frequency range, electrical charge and voltage level to be harvested. 

### 3.2. Further Optimization of the Piezoelectric Energy Harvester

The piezoelectric energy harvester was further optimized by replacing the steel ball (7.86 g/cm) with a tungsten (19.3 g/cm³) mass (Figure 10). The more than two times higher density of the tungsten ensured that the size of a tip mass did not increase, even when more than doubling the weight. A heavier tungsten mass ensured a lower resonance frequency even with a slightly thicker steel (250 µm), which in turn made it possible to transfers energy into a slightly thicker piezoelectric material (0.6 mm). Additionally, the piezoelectric (PZ-5A) area (25 mm × 30 mm × 0.6 mm) was increased by 30% compared to previous prototypes to further enhance the harvested energy.

The highest calculated raw power of 579.2 µW at 7.4 Hz rotational frequency was measured with Proto 1 in Figure 10. This computes to ~1287 µW/cm³ for piezoelectric material. This average power was around the same as that measured by Zou et al. in [24] (7.0 Hz/564 µW and 9.2 Hz/535 µW) from a two-beam revolving cantilever system. In Zou’s system, energy harvesting was enhanced by placing magnets on the tips of cantilevers pointing inward and facing each other. Figure 11 shows results of the prototypes’ harvested power as a function of rotating frequency and calculated days needed to overcome a 220 mAh (typical lithium CR2032 battery) with four harvesters. With this power level, a typical CR2032-battery could be loaded inside 6 days using four Proto 1 type piezoelectric cantilevers. According to the results, both prototypes 1 and 2 could theoretically support an electrical system if the rotational speeds were to be around 4 to 6.5 Hz and the electrical system functioned with a 220 mAh battery over a hundred days. Theoretically, the battery could be replaced with the introduced energy piezoelectric cantilevers if the battery life of a system exceeds the limits in operation time shown in Figure 11. Proto 2 with two tungsten masses (56.6 g) also produced close to the same power levels as Proto 1, even when its tip mass movement was more restricted with 3 mm cushions (Figure 10, red circles). The Proto 2 tip mass became more easily pinned/trapped into a stagnant position (6.5 Hz) although the power curves were quite similar. This is probably due to the centrifugal force being increased with the heavier tip mass.

## 4. Conclusions

In industry, anticipating machinery failures could save significant time and money if maintenance breaks are predictable. In this work, several piezoelectric cantilever-type energy harvesters were realized and measured from a rotating source. Pointing the tip mass inward decreases the harvesting beam resonance frequency and enhances the harvested energy significantly. Furthermore, restricting the movement of the tip mass can be used to harvest energy from a wider frequency range and reduce bouncing as well as avoiding getting trapped into a stagnant position. Correct balance of the piezoelectric material and steel thickness/length/width, tip mass and restricting the movement can be designed when the rotation frequency range of a system is known. The highest recorded continuous power was 579.2 µW at 7.4 Hz rotational frequency from one piezoelectric bimorph-type cantilever. Results suggest that a possible multibeam structure could replace batteries in the condition sensors monitoring of revolving machinery when rotational speeds are known, and an energy harvesting system can be designed accordingly together with optimal energy harvesting electronics. Further improvements could be made where the beam movement is restricted by other beams and tip masses, which could collide and force each other away from a stagnant position. This would enable the harvester to operate over a wider frequency range.

## Figures and Tables

**Figure 1 sensors-22-07449-f001:**
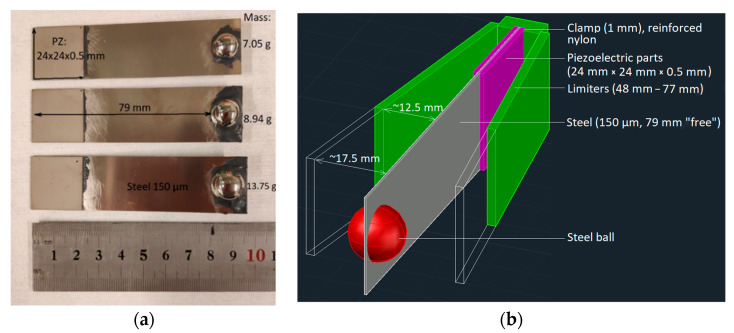
Piezoelectric bimorph type cantilever energy harvester prototype (**a**) dimensions and (**b**) a schematic of an energy harvester.

**Figure 2 sensors-22-07449-f002:**
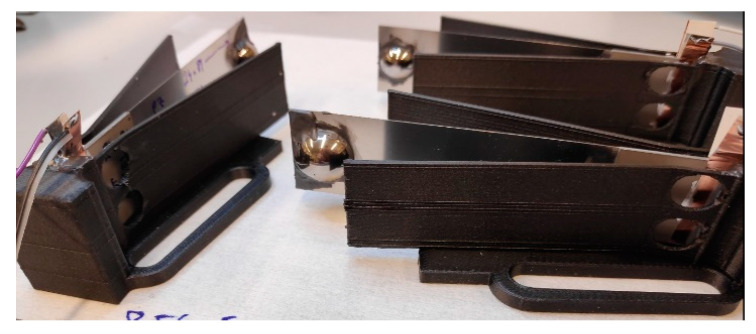
Prototype cantilevers clamped into nylon-reinforced 3D-printed holsters.

**Figure 3 sensors-22-07449-f003:**
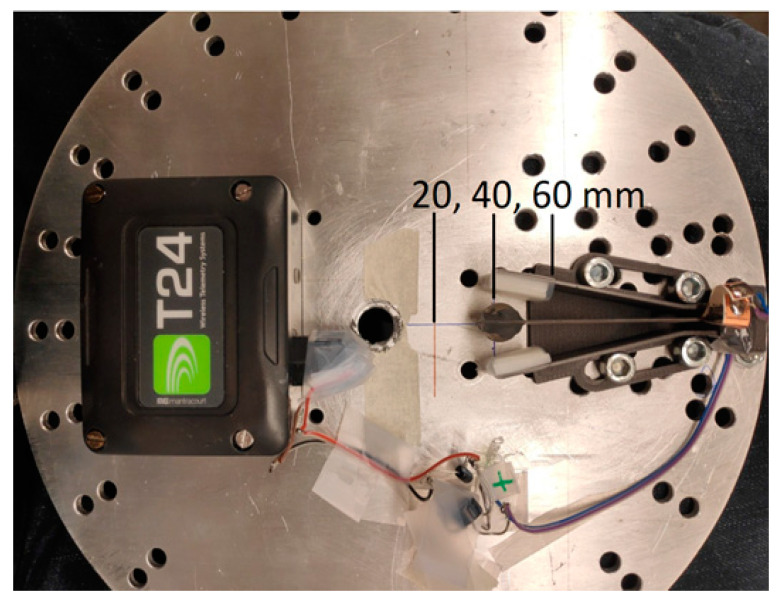
Alignment of piezoelectric cantilever harvester. Tip mass pointing toward center axis.

**Figure 4 sensors-22-07449-f004:**
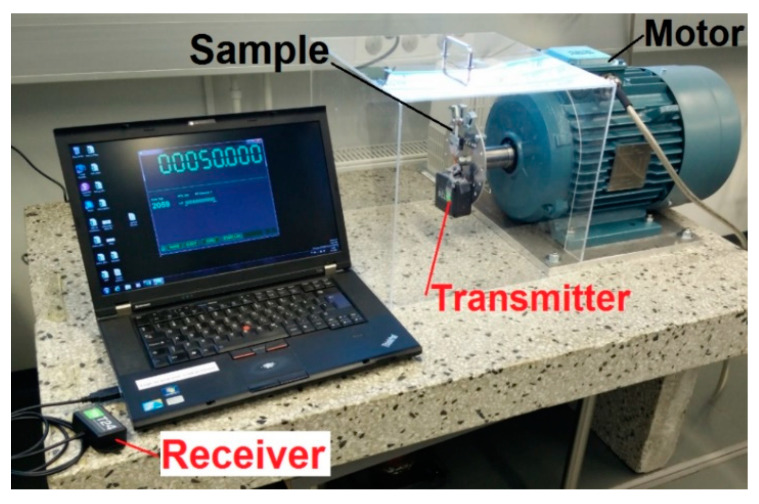
Measurement setup.

**Figure 5 sensors-22-07449-f005:**
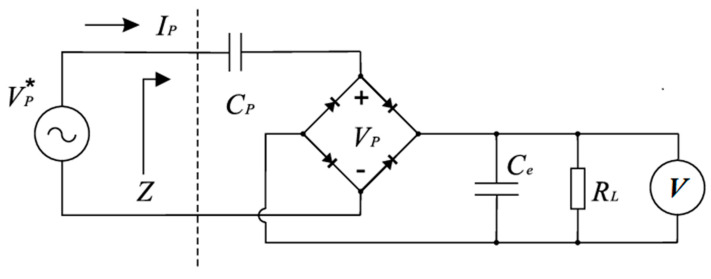
Energy harvesting circuit.

**Figure 6 sensors-22-07449-f006:**
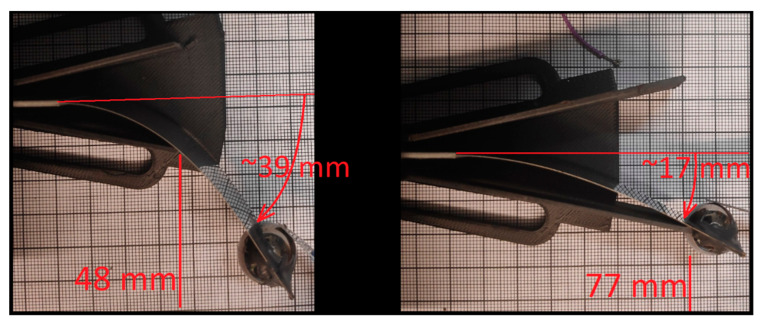
Tip mass movement with different limiter.

**Figure 7 sensors-22-07449-f007:**
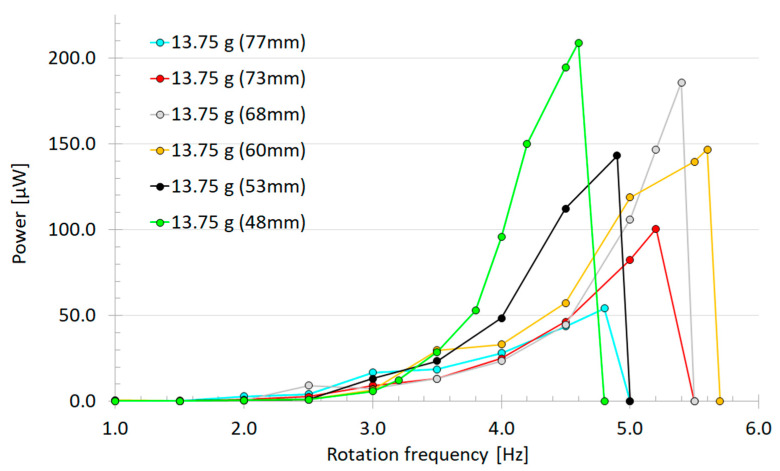
Harvested power as function of rotation frequency of the prototype with a 13.75 g tip mass pointed toward the center axis and measured with adjusted tip movement limiters from 48 mm to 77 mm.

**Figure 8 sensors-22-07449-f008:**
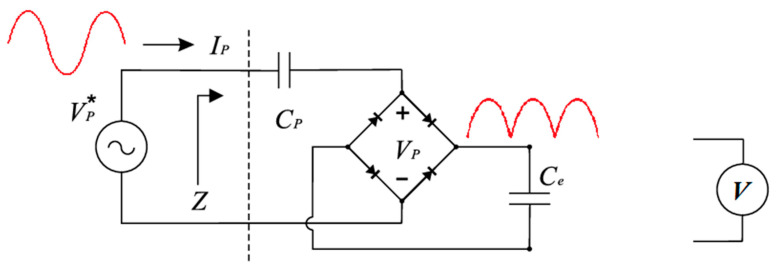
Energy was harvested through a rectifier into a capacitor.

**Figure 9 sensors-22-07449-f009:**
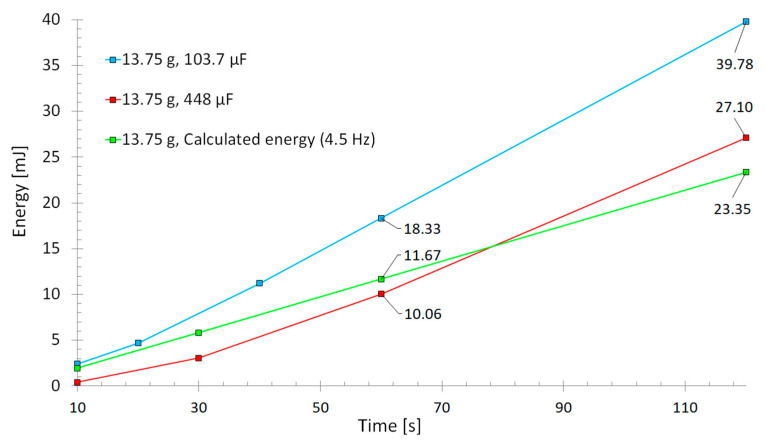
Harvesting electronics comparison. Blue and red lines indicate energy from capacitors loaded trough rectifier. Green line is calculated energy from measured voltage through a rectifier and over a load.

**Figure 10 sensors-22-07449-f010:**
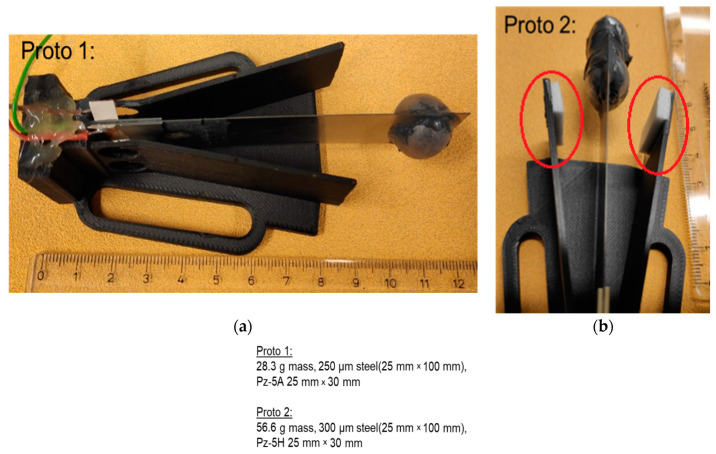
Piezoelectric cantilever prototypes with (**a**) a 28.3 g and (**b**) a 56.6 g tungsten mass.

**Figure 11 sensors-22-07449-f011:**
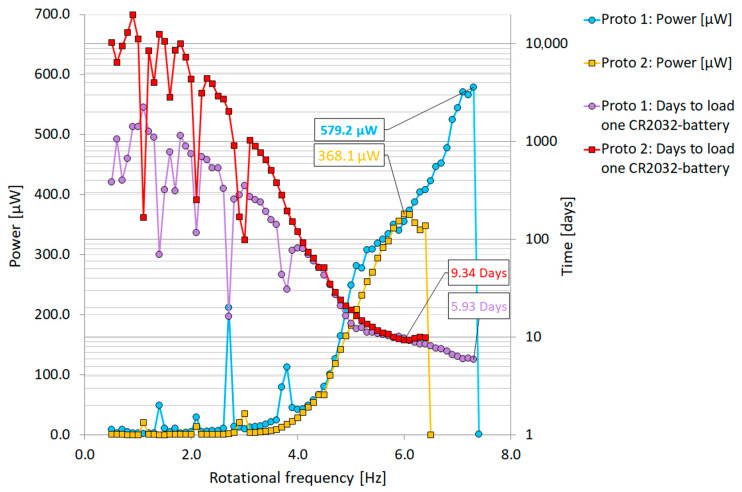
Piezoelectric energy harvesting cantilevers with a tungsten mass. Power as function of rotation frequency and calculated days to overcome a 220 mAh battery.

## Data Availability

Not applicable.

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
