# Peer review of "Piezoelectric Energy Harvesting from Rotational Motion to Power Industrial Maintenance Sensors"

_sensors, 2022, doi:10.3390/s22197449_

Round 1

Reviewer 1 Report

Can authors please provide more information on how this energy harvester is targeted to be used and applied? It says rotational motion but from where?  Where is it applied to?

There is no information on the mechanical schematic of the EH and what are the mechanics behind it.  What are the essential parameters that govern the output?  Is there any model to describe the EH?

The plots are poorly formatted.

The power of W/volume would be more relevant.

Author Response

Hi,

Thank You for constructive suggestions to improve the article.

We have made changes according to suggestions by four reviewers:

Major changes were made to organize the material and methods to separate sections and results to own section. More insight was added to introduction and motivation to this work. Also details to construction to energy harvesting mechanics and working principle. Separate paragraph to harvesting electronics. Many new citations added to give more insight to introduction, electronics and comparison to other works. Language checked again.

- Added more insight to harvested energy use cases into introduction as You suggested.

- Added a schematic figure (3b) of the energy harvester. 

- Added insight to essential parameters that govern the output of harvested energy amount.

- I don't have any model to descibe the EH.

- Hopefully plots are easier to read with added information.

- Added µW/cm^3 calculations (power vs PZ-volume).

Reviewer 2 Report

 Reviewer _ Sensors-1897591

Title: “Piezoelectric energy harvesting from rotational motion by 2 pointing cantilever towards centre axis”

Reviewer: The author should improve the title, this reviewer suggest something broader, less specific and that is also somehow "marketable" (that sounds more inviting)

Abstract

Reviewer: The abstract, in general, should have a more organized structure, as if it were the reduced manuscript, with a miniintroduction, development, results and conclusions described in a few words. This reviewer understands that the abstract presented is well reported, but the text would be more appropriate in the conclusion.

Keywords: “energy harvest; piezoelectric; rotational motion; softening effect” Reviewer: The author should choose keywords that do not appear in the title, because the title itself already provides keywords, so I strongly suggest that the keywords are different from those that appear in the title (remember this when changing the title).

1. Introduction

Reviewer:

1) The author should format the work according to “Instructions for Authors” in the “Submission Guidelines”;

2) This topic is much discussed in the literature, there are many studies on energy harvesting that were not covered in this work and that bring a bibliographic base that left something to be desired. This reviewer suggests the insertion of a bibliographic review section or even a chapter 1 that covers in a more comprehensive way the state of the art of the presented study.

2. Prototype manufacturing and test setup

Reviewer:

1) This reviewer considers that the paragraphs are too long, but that they could be broken, and interspersed with the figures as they are mentioned.

2) If section 2 is a materials and methods section, this should be modified. Present the materials first and then explain the methodology as it is all mixed up.

3) Take the results from this section and create a specific section for this.

Testing different harvesting electronics.

Reviewer:

1) separate the unit from the number, express the equations outside the text, number them and quote them in the text.

2) Bring result to result section (which doesn't exist yet);

3) Bring materials and methods to a materials and methods section (which doesn't exist yet).

Further optimization of the piezoelectric energy harvester

Reviewer:

1) Bring result to result section (which doesn't exist yet);

2) Bring materials and methods to a materials and methods section (which doesn't exist yet).

4. Discussion

Reviewer:

There is no section number 3, from 2 it goes to 4, but in the middle, there are two sections with no number... When it improves the organization of the presentation of the work, I will be able to pay more attention to other points that also deserve to be reviewed.

References

Reviewer:

1) Standardize references according to the model of this Journal.

2) Insert more references. The subject of this work is widely discussed in the literature; therefore the author should substantiate it better.

3) The author should try to include at least one reference from this Journal.

More comments in the revised article archive.

Author Response

Hi,

Thank You for fast response and insightful suggestions to improve the article.

We have made changes according to suggestions by four reviewers:

Major changes were made to organize the material and methods to separate sections and results to own section. More insight was added to introduction and motivation to this work. Also details to construction to energy harvesting mechanics and working principle. Separate paragraph to harvesting electronics. Many new citations added to give more insight to introduction, electronics and comparison to other works. Language checked again.

- TITLE: changed the title to focus more on industrial aspect as the work it self was also done partly for industrial partners.

- ABSTRACT: Added motivation and targets behind this research.

  • KEYWORDS: title change takes away some of the words in keywords and I also added "condition monitoring".

INTRODUCTION

  • 1) I used the Sensors template to create the manuscript. Some small figure errors/formats corrected.
  • 2) Added more citations widen bibliographic base. Included energy harvesting circuit related citations but also highlighted that this work was concentrated on optimizing and to studying mechanics behind the cantilever softening effect and movement limiter effects. Added also some specific wind induced energy harvesting articles that were missing. Added a comparison to harvested power by Zou et al. article which had similar revolving speeds.

2. Prototype manufacturing and test setup

  1. Figure moved to suggested locations.
  2. Materials and manufacturing explained first and methdology next as suggested.
  3. created a different section for results and subsections for "different harvester electronics" and "further optimization".

Testing different harvesting electronics.

  1. Equations separated from text.
  2. Materials moved to material section.
  3. Results moved to result section, methods moved to method section.

4. Discussion

Sections numbered from again and reorganized.

References

1)  References Standardized according to the model of this Journal.(Track changes not applied)

2) More references inserted and discussed more in the text.

3) Reference included from Sensors.

Reviewer 3 Report

The paper "Piezoelectric energy harvesting from rotational motion by pointing cantilever towards centre axis" is in conference format both in terms of writing and scientific contribution. I recommend a better coverage of the references. Much more details about the experiment and the materials (PZT!?) used so that it can be reproduced by interested readers. Is the load chosen for measuring power optimal? The work in the ensemble is well thought out and organized, but I think a slightly expanded version would bring more clarity. An approach to the compartment of the deformable structure (cantilever) from the perspective of the frequency response is welcome. The introduction of limiters has strong effects on its behavior. The work lacks a characterization of the deformed structure from the perspective of its dynamics. At this stage, the work is devoid of any theoretical approach, simulation, .., only experimental results are presented without having a referent in the literature for a possible comparison. The paper has no conclusions, an additional proof that it is not yet in a publishable form.

Author Response

Hi,

Thank You for the comments to improve this study.

We have made changes according to suggestions by four reviewers:

Major changes were made to organize the material and methods to separate sections and results to own section. More insight was added to introduction and motivation to this work. Also details to construction to energy harvesting mechanics and working principle. Separate paragraph to harvesting electronics. Many new citations added to give more insight to introduction, electronics and comparison to other works. Language checked again.

Manuscript has been modified to give more insight to what were goals and motivation behind this research. Motivation coming from industry maintenance and predicting machinery failures.

I Also  this work goal was mainly to focus on optimize and study mechanical 

  • References have been added to improve wider coverage piezoelectric energy harvesting generally and also from harvesting circuit point of view.
  • Materials have been listed with more detail and a schematic(Figure 3b) of piezoelectric(PZ-5A) bimorph type cantilver harvester has been added.
  • Electrical load was determined by wireless voltage measurement device(not optimal by far) and also the used harvester electronics were not optimal. These were the reasons for measuring also by directly loading a capacitor for comparison. Altough this study purpose was mainly focused on mechanical study, I wanted to stress the importance that the mechanical and electrical designing goes hand in hand for an optimal energy harvesting device.
  • More insight has been added to main essential parameters that govern the output of harvested energy amount.

Reviewer 4 Report

1. The energy harvesting circuit in Figure 8 is not optimized, and its own power loss is large. Please refer to advanced circuits, such as hybrid rectifier topology with self-starting up capability in [1-2],

[1] C. Lu, C. -Y. Tsui and W. -H. Ki, "Vibration Energy Scavenging System With Maximum Power Tracking for Micropower Applications," in IEEE Transactions on Very Large Scale Integration (VLSI) Systems, vol. 19, no. 11, pp. 2109-2119, Nov. 2011, doi: 10.1109/TVLSI.2010.2069574.

[2] C. Lu, C. -Y. Tsui and W. -H. Ki, "Vibration energy scavenging and management for ultra low power applications", Proceedings of the 2007 international symposium on Low power electronics and design, pp. 316-321, 2007.

[3] Minseob Shim, Jungmoon Kim, Junwon Jeong, Sejin Park, Chulwoo Kim, "Self-Powered 30 µW to 10 mW Piezoelectric Energy Harvesting System With 9.09 ms/V Maximum Power Point Tracking Time", IEEE Journal of Solid-State Circuits, vol.50, no.10, pp.2367-2379, 2015.

[4] X. -D. Do, H. -H. Nguyen, S. -K. Han, D. S. Ha and S. -G. Lee, "A Self-Powered High-Efficiency Rectifier With Automatic Resetting of Transducer Capacitance in Piezoelectric Energy Harvesting Systems," in IEEE Transactions on Very Large Scale Integration (VLSI) Systems, vol. 23, no. 3, pp. 444-453, March 2015, doi: 10.1109/TVLSI.2014.2312532.

2. Maximum power point tracking (MPPT) is critical for piezoelectric energy harvesting. You should briefly review this issue and compare with your work. How do you do the MPPT?

3. You should list a table and comprehensively compare your piezoelectric energy harvesting systems with other works. Then, the contribution and novelty of this work is clear.

4. Please summarize the contribution of this work in the abstract section.

Author Response

Hi,

Thank You for insightful suggestions to improve this article.

We have made changes according to suggestions by four reviewers:

Major changes were made to organize the material and methods to separate sections and results to own section. More insight was added to introduction and motivation to this work. Also details to construction to energy harvesting mechanics and working principle. Separate paragraph to harvesting electronics. Many new citations added to give more insight to introduction, electronics and comparison to other works. Language checked again.

  1. You are correct that the used energy harvesting electronics is far from optimal. I added Your citations examples to the article and emphasize the fact that this work was done mainly to improve and study the mechanics behind energy harvesting.
  2. I'm not an expert on circuit design and I further added to text that mechanical design should go hand in hand with electrical design to achieve best possible outcome.
  3. Comparing other works in a chart is always a bit harsh and shallow as designs are always made for a unique purpose. Some studies may solely focus on electronic performance optimization(low frequency, high frequency, wide frequency range, low input levels, high voltage input, random input, harmonic input... optimization) Same goes with mechanical designs optimization(high freq, low freq,..) Energy harvester in an application approach such as this study You also have to consider long term robustness and shield from sudden harmfull ambient forces.
  4. Abstract and title has been modified to give more insight to what were goals and motivation behind this research.

Round 2

Reviewer 1 Report

The authors have not address the previous comments.

1. Power should be power/volume.

2. Poorly formatted plots.

In addition,

1. Equation 1, the energy in a capacitor is E=1/2 CV2 and is not CV.  Please revisit into your energy calculation and related results.

2. It lacks numerical treatment to validate their observation. Further, were there any stress analysis to understand the fatigue of the piezomaterials to determine the operating limits?

Author Response

  1. Power should be power/volume.

- Already added "1287 µW/cm³ for piezoelectric material" for the further improved harvester. Added power/volume to the first prototype version with 13.75g/Ø 15 mm tip mass:  "~725 µW/cm³ for piezoelectric material."

2. Poorly formatted plots.

- Formatted plots 7, 9 & 11: changed fonts bigger, altered colors and text placement to make plots more clear.

  1. Equation 1, the energy in a capacitor is E=1/2 CVand is not CV.  Please revisit into your energy calculation and related results.

- I don't know how this error was applied. Calculations are correct but the formula was changed from original manuscript. It is now changed back to original.

2. It lacks numerical treatment to validate their observation. Further, were there any stress analysis to understand the fatigue of the piezomaterials to determine the operating limits?

- Added more numerical data into text from plots 7,9&11. Unfortunately no stress analysis was made for this prototypes. Only that they all were intact after measurements. It seems that movement limiters prevent the cantilevers from bending too much and also eliminate any sharp collisions.

Reviewer 2 Report

Dear authors,

                 The revision was successfully incorporated, however, there is still a lot of structural improvement work to be done,

In order to help make it more organized, I left some additional comments.

Author Response

  • Added wording "condition monitoring" also into the introduction. Discussion already had these words.
  • moved citation brackets into right places.
  • moved figures correctly after called in the text.
  • In the sensor journal template figure text a) b) have been inserted as a text.
  • Sections 3 & 4 renamed.
  • Equations moved into methodology section.
  • Separated figure 10 from figure 11 with text.
  • Made changes to plot figures 7,9 & 11: larger text and numbers, more vivid colors and markers, removed caption in figure 7 & 9 under the figure.

Reviewer 3 Report

In the revision version of the paper “Piezoelectric energy harvesting from rotational motion to 2

power industrial maintenance sensors” the authors improve the original article formally.  From certain points of view that will be described below, we are dealing with a progress in the opposite direction to expectations.   

E1. Equation (1) is wrong E = CV2/2. The energy stored in the capacitor is representative of your system if the reactance of this capacitor is equal to the reactance of the static capacitance plus the dynamics capacitance of the PZT at the working frequency. After the voltage at the capacitor electrodes is equal to the voltage from PZT, the charging process no longer occurs.  The process is described by an exponential function over time.

22. The fact that the load resistance is not optimal – that is, the value of the resistor to be equal to the reactance of the static and dynamic capacitance of the PZT to the working frequency  - makes that the reported power does not bring a scientific information that can be considered representative for the system proposed by you. A power response curve delivered by the PZT vs. the load resistance described by Equation (3) reveals a maxim in accordance with the condition set out above. This is the maximum power and this provides scientific information about the possible benefit of the proposed experimental setup (not a particular situation).

33. The transfer of energy from a mechanical structure to a PZT element is maximum in resonant conditions, a condition in which the coupling factor is maximum.  Of course, no optimal conditions are required in the exploitation regime and we can put as a load any arbitrary load but scientifically we must know the performance of the system in optimal conditions in order to prove the novelty element of your investigation.

44. The resonant frequency of the system is very low and the system is not exploited in resonant conditions and it is well known that a PEH has a very low efficiency at low frequencies where its exploitation is not recommended. That is why there are in literature various configurations of the deformable structure that allow an increase in the frequency of resonance in the system. That enormous mass attached greatly decreases the resonant frequency of the cantilever. Moreover, the system is designed to work in conditions of nonlinearity. Nonlinear PEHs are a special case in the literature at the beginning of the investigation process in relation to the multitude of situations that can be described theoretically or experimentally validated.

55. A study of the behavior of the deformable mechanical structure (the frequency response of the cantilever) can bring the necessary elements to highlight an eventual advantage of the proposed experimental setup.

1.                   Everything I write myself is described in thousands of articles related to this topic. An example of a particular case insufficiently investigated and outside of well-established rules in literature in this field cannot be considered as a scientific contribution.

66. Out of respect for your work to imagine and realize your experimental configuration, I recommend a major new review that not only formally covers the previous guidelines, but also brings experimental elements and evidence to the meaning of what is suggested.

Author Response

E1. Equation (1) is wrong E = CV2/2. 

-The calculations are correct and also equation was correct in the first manuscript version. Equation is now changed equation back to original.

22.  

Your point that the lack of optimal resistance measurements makes mechanical design comparison difficult is correct. But according to my research and experience, it does not make it impossible:

-If a capacitance of piezoelectric material, electrical load and harvesting frequency are kept close to same in the improved PEH version it can be compared to previous versions.

-Also this research did not focus entirely on harvested energy amount but also frequency and bandwidth control by the movement limiters.

Furthermore optimal resistance would always change with harvesting frequency, piezoelectric material capacitance and with harvesting electronics. In a partly nonlinear(mostly linear in this case) system recording each frequency, with different optimal electrical load, for each PZ-material amount and with each movement limiter length would be an insurmountable work.

I hope You can see some scientific value in this study.

Sincerely 

Jaakko Palosaari

Reviewer 4 Report

The quality of this manuscript is improved a lot.

Author Response

Thank You for encouragements!

I have made further improvements on the plots(fig.7,9,11) by color and font changes. Some structural changes by placing formulas into methodology section. Also some minor figure placing and data/word additions. 

Round 3

Reviewer 1 Report

I am happy with the revision.

Reviewer 3 Report

 In the paper "Piezoelectric energy harvesting from rotational motion to power industrial maintenance sensors" the authors have made a progress both from the perspective of the presentation of the application and the motivation of choosing deformable mechanical structure in particular operating conditions. It is difficult and expensive to take a standard approach to evaluating the parameters of the system investigated from a mechanical or electrical perspective. The work is interesting and should be regarded as a preliminary investigation with proven results from the perspective of a particular application. In this form I consider that the work meets the conditions of publication.